# A bottom-up emission estimate for the 2022 Nord Stream gas leak: derivation, simulations and evaluation

Rostislav Kouznetsov[1], Risto Hänninen[1], Andreas Uppstu[1], Evgeny Kadantsev[1], Yalda Fatahi[1], Marje Prank[1], Dmitrii Kouznetsov[2], Steffen Noe[3], Heikki Junninen[4], and Mikhail Sofiev[1]

[1]Finnish Meteorological Institute, Helsinki, Finland
[2]University of Electro-Communications, Tokyo, Japan
[3]Estonian University of Life Sciences, Tartu, Estonia
[4]University of Tartu, Estonia

**Correspondence:** Rostislav Kouznetsov (Rostislav.Kouznetsov@fmi.fi)

**Abstract.** A major release of methane from the Nord Stream pipelines occurred in the Baltic sea on 26 September 2022. Elevated levels of methane were recorded at many observational sites in northern Europe. While it is relatively straightforward to estimate the total emitted amount from the incidents (around 330 kt of methane), the detailed vertical and temporal distributions of the releases are needed for numerical simulations of the incident. Based on information from public media and basic physical concepts, we reconstructed vertical profiles and temporal evolution of the methane releases from the broken pipes, and simulated subsequent transport of the released methane in the atmosphere. The parameterisation for the initial raise of the buoyant methane plume has been validated with a set of large-eddy simulations by means of UCLALES model. The estimated emission source was used to simulate the dispersion of the gas plume with the SILAM chemistry transport model. The simulated fields of the excess methane led to noticeable increase of concentrations at several carbon-monitoring stations in the Baltic Sea region. Comparison of the simulated and observed time series indicated an agreement within a couple of hours between timing of the plume arrival/departure at the stations with observed methane peaks. Comparison of absolute levels was quite uncertain. At most of the stations the magnitude of the observed and modelled peaks was comparable with natural variability of methane concentrations. The magnitude of peaks at a few stations close to the release was well above natural variability, however the magnitude of the peaks was very sensitive to minor uncertainties in the emission vertical profile and in the meteorology used to drive SILAM. The obtained emission inventory and the simulation results can be used for further analysis of the incident and its climate impact. They can also be used as a test case for atmospheric dispersion models.

## 1   Introduction

A major release of methane from the Nord Stream pipelines 1 and 2 occurred at the bottom of the Baltic sea on 26 September 2022 as a result of explosions at both lines. At the moments of the blasts, the pipes were filled with pressurized methane but no gas pumping was happening. Over the following days, methane escaped from the damaged pipes to the atmosphere.

Natural gas mining and transport through pipelines are considered among the safest means of energy transport. Over the period 1800 - 2018, less than 300 serious accidents have been documented worldwide, which is, for instance, 4 times less than

in oil transport, 8 times less than in the coal industry, and 10% lower than the accidents counted in wind energy (Kim et al., 2021). In the standard practice, accidents in the energy sector are categorized in terms of fatalities and property damage, which are documented by the authorities (e.g., Pipeline and Hazardous Materials Safety Administration in the US). Other parameters, such as the amount of natural gas released into the atmosphere, are rarely considered. However, in the Nord Stream case, the atmospheric release was one of important characteristics of the incident. To put it into a large-scale context, one can note the annual release of methane from the US gas production and distribution system was 13 Tg (+2.1/-1.6 Tg, 95% confidence interval) in 2015, i.e. 2.3% of the total production in that year (Alvarez et al., 2018). This number includes both releases from normal and abnormal operations and significantly exceeds the official US EPA methane emission of 8.1 Tg for 2015 (EPA, 2017). Alvarez et al. suggested that the disagreement is partly due to accidental releases, which are not accounted for in the official EPA inventory. They estimated the gas transportation-only contribution to the $CH_4$ total emission as 1.8 Tg/y (both normal and abnormal operations), whereas the US EPA regular-operation estimate is 1.4 Tg/y (normal operations). Comparison of these numbers suggests that the accidental losses in the US gas transport system are ∼400 Gg/y (EPA, 2017). The release from one of three breached Nord Stream pipes was estimated to be 115 Gg (Sanderson, 2022), i.e. over 30% of the above annual leaks due to accidents at the US pipelines (over 5 mln km of the total length), but accounts for around 0.14% of the global annual methane emissions from the oil and gas industry (Sanderson, 2022). Therefore, albeit extremely large for a single case, the Nord Stream leaks alone could hardly have a measurable impact on the global scale (Chen and Zhou, 2022).

The long atmospheric lifetime of methane and its significant radiative effect make methane a major greenhouse gas (Tollefson, 2022). Since it also has a very low deposition velocity and solubility (100 times less soluble in water than $CO_2$), its release at virtually any height leads to large-scale dispersion. Besides that, methane is flammable at mixing ratios of 5 to 15 volume percent (Zabetakis, 1964) and in large concentrations it can be very hazardous due to oxygen deprivation (e.g. Duncan, 2015). Therefore, emergency management of large releases, similar to the one considered in this study, requires detailed knowledge of the release temporal and vertical distribution, and evolution of the resulting in-air concentrations.

Methane density is about half of the air density, therefore a concentrated release of methane creates a powerful buoyant plume, which rises in the atmosphere similarly to an overheated plume from a major fire. Numerous (semi)empirical models and parameterisations have been developed for estimating the equilibrium height and vertical profile of atmospheric injection of buoyant plumes. However, these models were developed for industrial stacks and provide unrealistic results with very powerful buoyant releases or releases that take place over extended area (Sofiev et al., 2012; Li et al., 2023). More suitable models for such conditions have been developed for vegetation fires (Freitas et al., 2007; Sofiev et al., 2009, 2012; Rémy et al., 2017).

The accidents at the Nord Stream pipelines have been extensively reported in mass media and by various Internet resources. Many mutually-contradicting facts about the pipeline, leak locations and their intensity have been published. Even the locations and number of leaks have been specified differently by different sources (Fig 1).

There have been several publications analysing the gas releases from the pipes. The total amount of about 110 kt of methane per pipe (around 330 kt in total) can be calculated on the back of an envelope if one assumes an initial pressure of 105 bar inside the pipes, a final pressure of 7 bar, and knows the pipe dimensions.

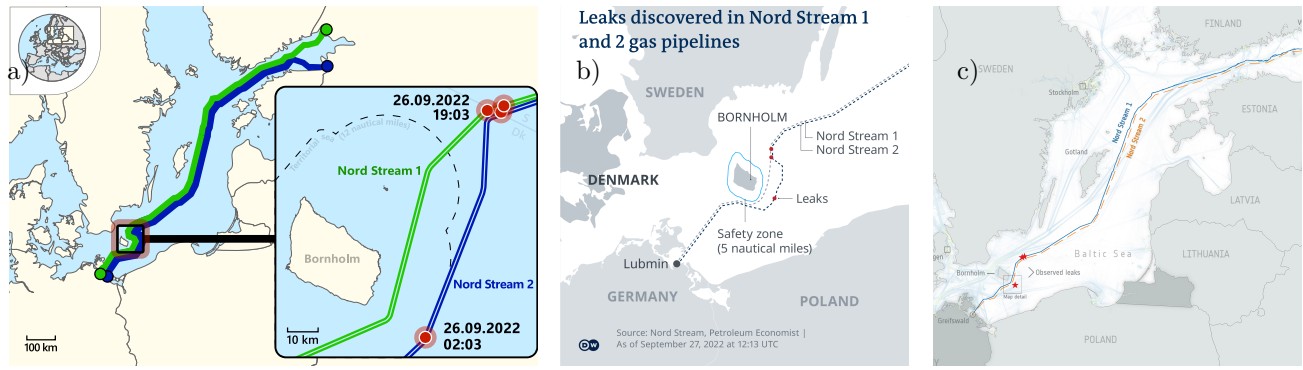

**Figure 1.** Maps of the Nord Stream gas leaks from various sources. a) Wikimedia https://upload.wikimedia.org/ wikipedia/commons/e/e1/Nord_Stream_gas_leaks_2022.svg (CC BY-SA 4.0), b) Deutsche Welle (https://www.dw.com/en/ denmark-sweden-view-nord-stream-pipeline-leaks-as-deliberate-actions/a-63251217), c) European Space Agency (ESA, https: //www.esa.int/ESA_Multimedia/Images/2022/10/Nordstream_pipeline_map_with_shipping_traffic).

The calculated amount varies depending on assumptions of the natural gas composition and the initial gas pressure. Such calculations have been performed in several studies. Jia et al. (2022) assumed that two pipes were destroyed, and thus reported 230 kt of total methane released. Sanderson (2022) reported 115 kt from the destruction of a single NS2 pipe. The worst-case

scenario considered by the Danish environmental agency amounts to 500 kt (https://ens.dk/en/press/possible-climate-effect-gas-leaks-nord-accessed on 15.8.2023), which is probably based on the design pressure of the pipeline, rather than on the actual pressure at the start of the release.

The total released amount has been analyzed also by inverse techniques. The Norwegian Institute for Air Research reported total emissions in the range between 56 and 155 kt of methane (https://www.nilu.com/2022/10/improved-estimates-of-nord-stream-leaks/,a

65 on 15.8.2023). Jia et al. (2022) reported $220\pm30$ kt, which nicely coincides with their bottom-up estimate. None of these studies considered the effect of methane buoyancy on the plume injection or initial release height.

The goals of the current paper are: (i) to construct a self-consistent and physically feasible picture of the event, (ii) to calculate the bottom-up time-resolving emission of methane from the broken Nord Stream pipes, (iii) to estimate the vertical extent of the plume injection and its evolution, (iv) to calculate the plume dispersion in the atmosphere during several days

since the release, by using the Finnish emergency and atmospheric composition model SILAM, (v) to evaluate the resulting simulations against observational data.

The paper is structured as follows. The next section describes the models and the observational data sets used to evaluate the inventory. Section 3 formulates a mathematical model for the temporal evolution of the leak intensity. Section 4 formulates an approach to evaluate the injection height for the buoyant methane plume. Section 5 summarizes the parameters of the pipelines

and the gas leaks that are available in the media and literature, and formulates the emission source for the Nord Stream 2022 gas leaks. It also compares the injection profile obtained from the parameterisation to the vertical distribution simulated with a

large eddy simulation model. Section 6 describes the simulations of the methane dispersion from the leaks and the results of a comparison against in situ observations of methane concentrations.

## 2 Modelling tools and measurement data

### 2.1 SILAM chemistry-transport model

To simulate the plume dispersion we have used the atmospheric chemistry transport model (CTM) SILAM (https://silam.fmi.fi). The model features a mass-conservative and non-diffusive Eulerian advection scheme (Sofiev et al., 2015), and has been used for many applications in the fields of research, operational forecasting and emergency-response. The model can operate at various scales, starting from sub-kilometer resolutions in a limited-area mode, to several-degree resolutions in a global mode. Feasible vertical resolutions normally start from around 10 m near the surface to several kilometer thick layers in free troposphere and stratosphere.

Being an offline CTM, SILAM requires a pre-computed set of meteorological fields to drive the transport and transformation processes. SILAM can consume meteorological fields from several different numerical weather-prediction models (NWP) and climate models. For the present study, we use high-resolution operational global forecasts (HRES product) by the European Center for Medium-Range Weather Forecasts (ECMWF), obtained with the Integrated Forecasting System (IFS), and forecasts from the unperturbed member of the Mesoscale Ensemble Prediction System (MEPS) for the Nordic countries. MEPS is based on the Harmonie meteorological model. From both models, a series of hourly forecasts with the shortest available lead time was used. The ECMWF forecast was taken with a resolution of $0.1° \times 0.1°$ degrees in a rotated lon-lat grid, and the MEPS forecasts were used at the original Lambert conformal conic grid of 2.5 km resolution. To evaluate the sensitivity of the simulations to the spatial resolution, we made three sets of the simulations: VHires at a $0.02° \times 0.02°$ grid, HiRes at a $0.1° \times 0.1°$ grid, and LoRes at a $0.4° \times 0.4°$ grid. The first of these was driven with data from the MEPS model, whereas the latter two were driven with the same set of data from the IFS model. All grids were aligned with the input meteorological grids. All simulations used the same vertical structure consisting of 13 stacked layers of increasing thicknesses, from 25 m at the surface to 2000 m close to the domain top, located at 6000 m above the surface.

SILAM allows for several types of meteorology-dependent emission sources, including a source for wildland fires with a dynamic injection height (Sofiev et al., 2012). For the current study, the fire plume-rise module has been interfaced to the point-source module, enabling injection of large buoyant plumes. For such sources, the buoyancy flux is provided along with the emission rate, and the former is used to evaluate the injection height range.

### 2.2 UCLALES large eddy simulator

The applicability of the fire plume rise module of SILAM for the current task was evaluated by comparing it to fine scale simulations of the buoyant plume made using the large eddy simulator UCLALES (Stevens et al., 1999, 2005; Stevens and Seifert, 2008). The methane emission was applied in UCLALES as a volumetric flux originating from the underlying surface.

The horizontal distribution of the emission flux was assumed to be normal, with 99% of the emission located in a circular area with a 500 m diameter. The formula for virtual temperature used for computing the vertical acceleration due to buoyancy was amended to account for methane mixing ratio in the grid cell.

UCLALES simulations were initialized with temperature, humidity, wind profiles, and surface variables taken from the same ECMWF forecasts used for SILAM simulations. Simulations were made in a 5 km high domain spanning 18 km in the downwind and 6 km in the crosswind direction. The domain was selected to be sufficiently large to extend beyond the vertical and cross-wind spread of the plume, and long enough in the downwind direction for the plume to rise to its final altitude. The simulations were made at a 50 m horizontal and a 10 m vertical resolution and a time step with a maximal length of 1 second, automatically reduced if required by the flow conditions for stability of the UCLALES numerical schemes.

## 2.3 Observational data

To validate our simulation results, we use observational time series of atmospheric methane concentrations obtained by the Integrated Carbon Observation System (ICOS) network (https://icos-cp.eu, accessed 12.12.2023). We use hourly time series of in-air volume mixing ratios of methane from several dozens of stations in Europe. Many stations are located in tall towers and are able to make observations at several different heights up to few hundred meters above the surface.

In the paper, we use data from five ICOS stations: the Finnish Utö station (UTO), located in the Baltic sea (Hatakka and Laurila, 2022), the Swedish Norunda (NOR) and Hyltemossa (HTM) stations (Lehner and Mölder, 2022; Heliasz and Biermann, 2022), and the Norwegian Birkenes (BIR) and Zeppelin (ZEP) stations (Lund Myhre et al., 2022a, b). Data from more ICOS stations are presented in the Supplementary material.

In addition to the ICOS stations, we have also used data from the Finnish measurement station at Sodankylä, FI-SDK (Kilkki et al., 2015), and two Estonian sites: the Järvselja SMEAR (Station for Measuring Ecosystem-Atmosphere Relations) EE-SMR (Noe et al., 2015), and the Tahkuse station THK (Luts et al., 2023; Hõrrak et al., 2000). Despite these stations using very similar protocols to the ICOS network, they are currently not a part of it.

In the figures below, the time series from the ICOS stations have been marked with their three-letter codes and the measurement heights above the ground in meters. The other stations have been marked with two-letter country codes and three-letter abbreviations of their names. The complete list of the stations, their locations and references for the ICOS time series used can be found in the Supplementary material.

## 3 Equations for a methane leak from a half-open pipe

To estimate the leak discharge as a function of time, let us consider an idealized system: a long smooth round pipe of inner diameter $D$ and length $L$ ($L \gg D$), which is closed at both ends and filled with a pressurized gas of initial density $\rho_0$. At the moment $t_0$ one end of the pipe is opened and the gas starts leaking.

The evolution of the gas velocity $v(x,t)$ and density $\rho(x,t)$ along the pipe can be described by the equation of motion:

$$\rho \frac{\partial v}{\partial t} = -\frac{\partial p}{\partial x} - \frac{\rho v^3}{2D|v|} f, \tag{1}$$

and the continuity equation:

$$\frac{\partial \rho}{\partial t} = -\frac{\partial (\rho v)}{\partial x}. \tag{2}$$

The first term at the right-hand side of Eq. 1 describes the acceleration of the gas due to the pressure gradient along the pipe and the second term describes the turbulent drag. The dimensionless drag coefficient $f$ depends on the flow regime. The relevant velocity for the flow ranges from about ten meters per second, to the speed of sound ($450\,\mathrm{m\,s^{-1}}$), and the kinematic viscosity of methane for the pressure range of $10 - 100$ bar can be approximated as $\nu = 15 \times 10^{-6}\,\mathrm{m\,s^{-2}} \cdot \frac{\rho_a}{\rho}$, where $\rho_a$ is the methane density at standard conditions. The Reynolds number $Re = vD/\nu$ of the flow exceeds $10^6$, but is less than $4 \times 10^7$. At such Reynolds numbers the Blasius formula for the drag coefficient is applicable:

$$f = (100\,Re)^{-1/4}. \tag{3}$$

To get a complete system of equations for $\rho(x,t)$ and $v(x,t)$, one needs also an equation of state that connects the pressure and density of the gas. Since the pipe is submerged in water, the process of gas expansion can be considered isothermal at temperature $T = 278\,\mathrm{K}$ (Kniebusch et al., 2019). The ideal-gas equation of state

$$p(x,t) = \frac{R}{\mu} \rho(x,t) T, \tag{4}$$

where $R$ is the universal gas constant ($R \simeq 8.3\,\mathrm{J\,K^{-1}\,mol^{-1}}$), and $\mu$ is the molar mass of the gas ($\mu = 0.016\,\mathrm{kg\,mol^{-1}}$), does not describe methane in the relevant pressure range. In particular, it predicts the density of methane at 100 bar to be some 20% lower than experimental values reported by Mollerup (1985). Therefore, we use the more rigorous van der Waals equation:

$$p(x,t) = \frac{R}{\mu} \left( \frac{1}{\rho(x,t)} - \frac{b}{\mu} \right)^{-1} T - \frac{a}{\mu^2} \rho^2(x,t), \tag{5}$$

where $a$ and $b$ are gas-specific van der Waals constants, describing the effects of the finite volume of a gas molecule, and the effects of inter-molecular attraction. For the study we use the values of $a = 0.21\,\mathrm{J\,m^3\,mol^{-2}}$, $b = 4.31 \times 10^{-5}\,\mathrm{m^3\,mol^{-1}}$. Note that our value of $a$ differs from the one suggested by Poling et al. (2001) ($a = 0.2303\,\mathrm{J\,m^3\,mol^{-2}}$), since it fits better experimental data on methane density, e.g. by Mollerup (1985), for pressures up to $150\,\mathrm{bar}$.

The initial and boundary conditions corresponding to the pipe are

$$\rho(x,0) = \rho_0, \tag{6}$$

$$v(x,0) = 0, \tag{7}$$

$$v(0,t) = 0, \tag{8}$$

$$\rho(L,t) = \rho_{\mathrm{out}}, \tag{9}$$

where $\rho_0$ is the density of the gas at the initial pressure $p_0$ (inside the pipe) and $\rho_{\text{out}}$ is the density at the pressure at the open end of the pipe, i.e. $p_{\text{out}}$.

Breaching a pressurized pipe at an intermediate point is equivalent to the simultaneous opening of the ends of two shorter pipes located at both sides of the breach.

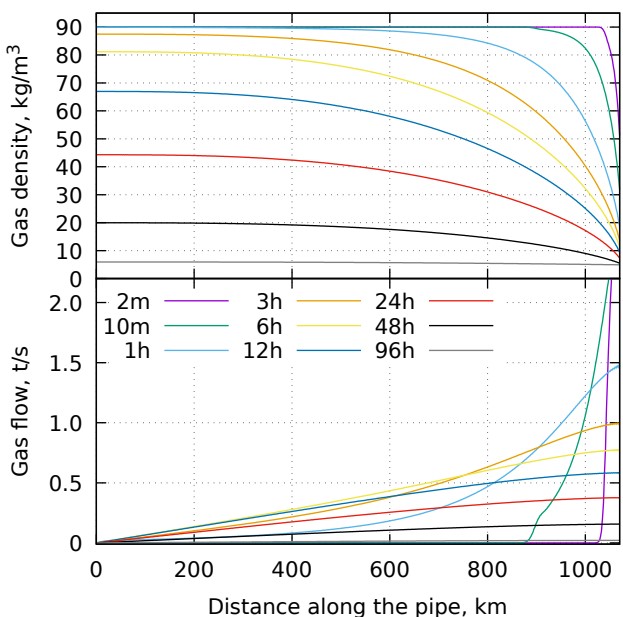

**Figure 2.** The along-pipe profiles of the density (upper panel) and the flow rate (lower panel) for different times (in minutes/hours, as indicated in the legend) after opening the pipe end for the pipe length of 1080 km.

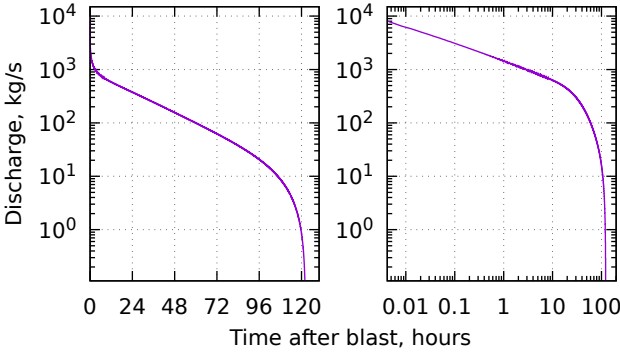

**Figure 3.** The gas discharge rate as a function of time after opening the pipe end for the same case as in Fig. 2 with linear (left) and logarithmic (right) time axis.

To illustrate the temporal evolution of the gas distribution within a pipe containing pressurized methane after one end has been opened, we consider a 1080 km long pipe of a an inner diameter of 1.153 m at the initial pressure of 105 bar and an outside pressure of 7 bar. Figure 2 shows the simulated profiles of the gas density and the gas flow along the pipe at various times. The evolution of the flow in the pipe has two stages. During the first stage the distortion propagates towards the closed end of the pipe, and during the second stage the flow profile is almost linear along the pipe, as the speed is limited by the turbulent drag inside the pipe. These regimes are clearly seen also in the evolution of the discharge rate at the open end of the pipe (Fig.3). Initially, the flow can be described by a power function of time, and the it starts to decay exponentially. Once the flow becomes so slow that the drag is insignificant, it ceases quickly.

## 4 Injection height

Methane is almost half as light as air. A massive injection of methane from the surface of the Earth produces a buoyant plume that rises upwards and is mixed with surrounding air, eventually losing its buoyancy. A number of various models and parameterisations have been developed for describing the buoyant plume rise from industrial sources (e.g. Briggs, 1984). However, these empirical formulas turned out to be inaccurate for highly-buoyant wide-area sources, for which alternative solutions were proposed. In particular, a dedicated semi-empirical parameterisation was suggested and evaluated for plumes from vegetation fires by Sofiev et al. (2012). Input variables for that approach are derived in this section.

The primary characteristic of buoyant plumes in plume-rise parameterisations is the buoyancy flux (Venkatram and Wyngaard, 1988, eq. 3.11 there):

$$F_{\mathrm{b}} = F_{\mathrm{v}} g \frac{\Delta \rho}{\rho_{\mathrm{a}}}, \tag{10}$$

where $F_{\mathrm{v}}$ is the volumetric flux of a source (in $\mathrm{m^3\,s^{-1}}$), $g$ is the acceleration of gravity, $\Delta \rho$ is the difference between ambient air density $\rho_{\mathrm{a}} \simeq 1.2\,\mathrm{kg\,m^{-3}}$ and the released gas density $\rho_{\mathrm{g}} \simeq 0.69\,\mathrm{kg\,m^{-3}}$. It is straightforward to convert a methane discharge at the surface $F_{\mathrm{m}}$ (in $\mathrm{kg\,s^{-1}}$) to the buoyancy flux:

$$F_{\mathrm{b}} = F_{\mathrm{m}} g \frac{\Delta \rho}{\rho_{\mathrm{a}} \rho_{\mathrm{g}}}. \tag{11}$$

For methane in standard conditions, the conversion coefficient $g \frac{\Delta \rho}{\rho_{\mathrm{a}} \rho_{\mathrm{g}}} \simeq 6\,\mathrm{m^4\,kg^{-1}\,s^{-2}}$

The parameterisation for plume injection heights for wildland fires Sofiev et al. (2012) uses the Fire Radiative Power (FRP) of a fire as a measure of its intensity. According to Wooster et al. (2005), FRP constitutes about 20% of the total combustion energy, competing with the convective energy loss, latent heat release, and heat conduction into the soil. In the same work, the conduction of heat into the soil was suggested to consume barely 5% of the total energy, thus leaving 75% of the total combustion energy distributed between sensible and latent heat releases, the former being the dominant fraction. These estimates corroborate with some works (e.g. Kremens et al., 2012), but are rather conservative in comparison with others (e.g. Ferguson et al., 2000). Admitting significant uncertainties in the relation between FRP and convective power, all studies agree that they differ by a factor of a few times at most. Since the formula of Sofiev et al. (2012) involves the cubic root of the FRP, one can

assume that for a fire the fractions of the power spent for radiation and for creating buoyancy are approximately equal. Thus the equivalent of FRP to a gas leak can be expressed in terms of the buoyancy flux.

The buoyancy of a given volume of methane at temperature $T_0$ is equivalent to the buoyancy of the same volume of air at a temperature of $T_{\text{eff}}$:

$$T_{\text{eff}} = T_0 \frac{\rho_{\text{a}}}{\rho_{\text{g}}}. \tag{12}$$

$T_{\text{eff}}$ is analogous to the virtual temperature used for buoyancy calculations of water vapour. The power needed to produce the overheated air plume of the same buoyancy as the release of the gas is:

$$\text{FRP} = F_{\text{m}} c_{\text{p}} T_0 \left( \frac{\rho_{\text{a}}}{\rho_{\text{g}}} - 1 \right), \tag{13}$$

where $c_{\text{p}}$ is the specific heat capacity of air at constant pressure. At standard conditions, the conversion factor to get the equivalent of FRP for a methane release is $\sim 1.9 \times 10^5 \, \text{J} \, \text{kg}^{-1}$, which is more than two orders of magnitude smaller than the specific energy of the released gas if it was combusted ($5.6 \times 10^7 \, \text{J} \, \text{kg}^{-1}$).

Therefore, the injection of methane at the surface of the Earth at a rate of $1 \times 10^4 \, \text{kg} \, \text{s}^{-1}$, as in the beginning of the release shown in Fig 3, is equivalent to a fire emitting 2 GW of radiation. It is of the same order of magnitude as the most powerful fire considered by (Freitas et al., 2007), and much higher than any realistic industrial sources. A smoke plume from such a fire, depending on the weather conditions, can rise up to a few kilometres due to its own buoyancy (Sofiev et al., 2012).

## 5 Quantifying the emission source

### 5.1 Reported locations, and timelines of the leaks

The Nord Stream 1 and 2 pipelines consist of two pipes each. The locations of the leaks, reported by the Danish Marine Authority, are shown in Fig. 4ab. The locations of the Nord Stream 1 pipeline and a part of the Nord Stream 2 pipeline as reported by the EMODnet human activities database (https://www.emodnet-humanactivities.eu, last access 9.12.2022) are shown with solid lines. A part of the Nord Stream 2 pipeline, missing from the database, is sketched with a dashed line that connects the westmost point of NS2 pipeline given by the database, the leak site NS2A, and the destination point of the pipeline.

A blast at the Nord Stream 2 pipeline was detected by a seismograph of the Danish Geological Survey at Bornholm island at 02:03 CEST (00Z) on 26.09.2022, and similar data were reported by several seismic stations in the region (https://www.geus.dk/om-geus/nyheder/nyhedsarkiv/2022/sep/seismologi, last access 9.12.2022). Soon after that the Nord Stream 2 pipeline's operators saw a sudden pressure drop from 105 bar to 7 bar in one of the pipes, (Sanderson, 2022), and a Danish F-16 interceptor discovered a gas leak at the location of the seismic wave origin (NS2A in Fig. 4a). On the same day, the area around the location was closed by the Danish Marine Authorities for all types of vessels with the Navigational warning NW-230-22. The bubbling of the water surface at the location was observed to go on for several days after the blast by satellites and aircraft. On 01.10.2022 Danish Energy Agency reported that according to the Nord Stream

2 operator, the pressure in the damaged Nord Stream 2 pipe stabilized and the gas leakage from the pipe ceased (https: //apnews.com/article/russia-ukraine-putin-united-states-germany-business-afebd99d298ac72192acfeabfe384609)

A series of blasts at the Nord Stream 1 pipeline was detected by the same seismographs around 19:03 CEST (17Z) on 26.09.2022. According to the Navigational warning NW-235-22 issued by the Danish Marine Authorities, three leaks were dis-covered: NS1A, NS1B, and NS2X in Fig. 4ab. NS1A, and NS1B correspond to leaks in both of the pipes of the Nord Stream 1 pipeline, whereas the location NS2X corresponds to the Nord Stream 2 pipeline. The leaks NS1A and NS1B were recorded by several satellites and aircraft during several days following the blasts. The leaks ceased on 02.10.2022 (https://sverigesradio.se/ artikel/nord-stream-1-har-slutat-att-lacka-gas). However, we could not find any information on further detections of leaks at the NS2X site after the initial one. As the second of the two Nord Stream 2 pipes stayed intact (https://www.reuters.com/ business/energy/gazprom-lowers-pressure-undamaged-part-nord-stream-2-pipe-denmark-says-2022-10-05/), while the leak from the NS2A site continued long after 26.09.2022, we conclude that the leak at NS2X was probably a mistake in the is-sued warning.

The key input needed to evaluate the amount released from the pipelines is the initial pressure inside the pipes at the moment of the rupture. Besides the aforementioned pressure of 105 bar, we could find an image of a pressure gauge seen at the landfall facility of the Baltic Sea gas pipeline Nord Stream 2 in Lubmin, Germany, 19.09.2022 (reuters.com/business/energy/ gazprom-lowers-pressure-undamaged-part-nord-stream-2-pipe-denmark-says-2022-10-05) that indicates a pressure of 95 bar. Therefore, we suggest that the accuracy of the release estimates based on the available pressure figures should be around 10-15%.

**Table 1.** Nord-Stream gas pipe and blast parameters assumed for the simulations

| Parameter | Value, notes |
| --- | --- |
| Pipe inner diameter $D$ | 1.153 m [a] |
| Pipe Length $L$ | 1224 km [a] |
| Initial pressure $P_0$ | 105 bar [b] |
| Water pressure at the blast point | 7 bar [b] |
| NS2A leak | started 26.09.2022, 00Z, 54.877N,15.410E [c] |
| NS1A leak | started 26.09.2022, 17Z, 55.535N,15.698E [d] |
| NS1B leak | started 26.09.2022, 17Z, 55.557N,15.788E [d] |
| NS2X leak | started 26.09.2022, 17Z, 55.53N,15.6983E [d], assumed false detection |

[a] Nord Stream AG (2013)

[b] Sanderson (2022)

[c] Navigational warning NW-230-22 by Denmark marine authority https://nautiskinformation.soefartsstyrelsen.dk

[d] Navigational warning NW-235-22 by Denmark marine authority https://nautiskinformation.soefartsstyrelsen.dk

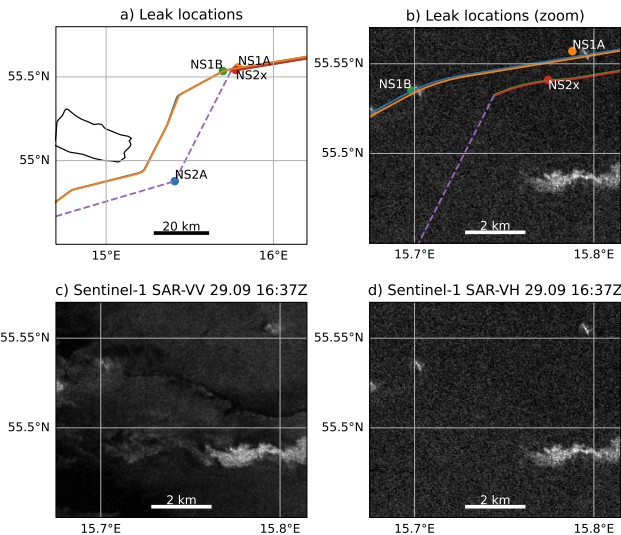

**Figure 4.** The locations of the gas leaks near the island of Bornholm reported by the Danish Marine Authority, and the Nord Stream pipeline (a), zoom plotted over the Sentinel-1 Synthetic-Aperture Radar backscatter acquired on 29.09.2022 at 16:36:54Z in VH polarization (b), and the same radar image in VV (c) and VH (d) polarizations. The lighter areas on the radar images indicate a disturbed water surface. The Sentinel-1 data were acquired from ESA via https://scihub.copernicus.eu.

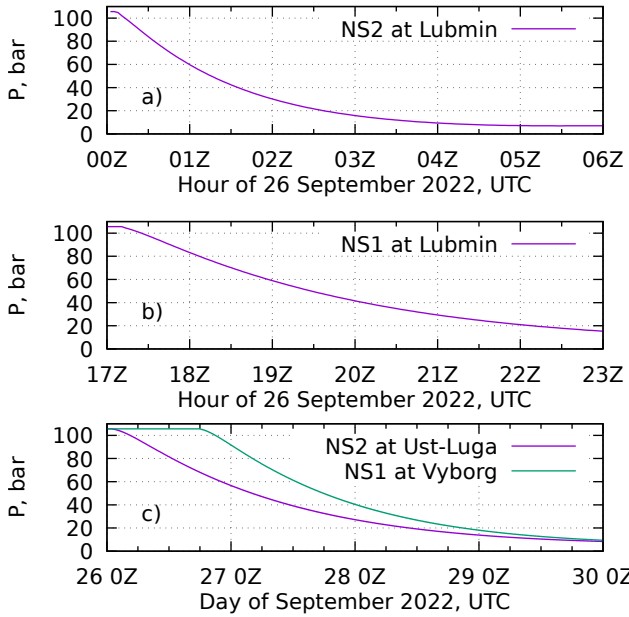

**Figure 5.** Pressure evolution at the landfall facilities of the Nord Stream pipelines during the leak events, according to our calculations. Note different time axes on the panels.

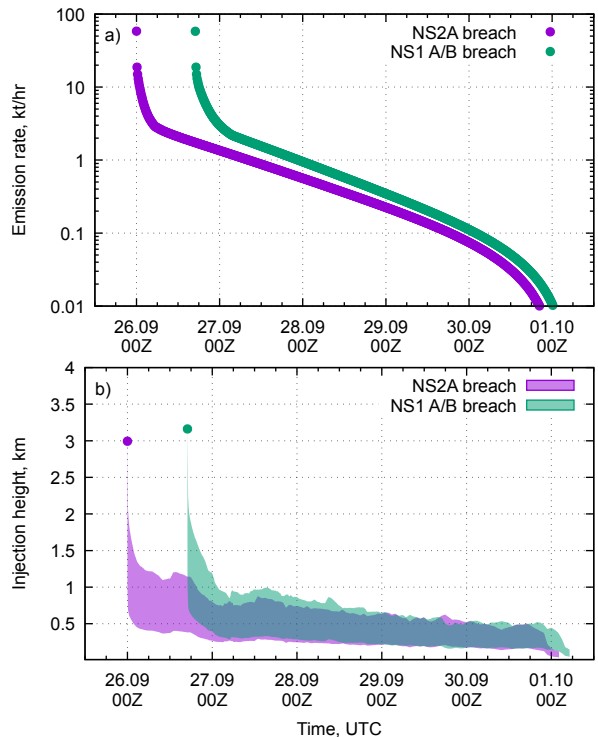

**Figure 6.** The simulated emission rates (a) and injection heights (b) from the breached pipelines.

## 5.2 The emission source

The gas discharge from each leak can be considered as a sum of two flows originating from half-opened pipes on both sides of the leak. For the NS2A leak, we take the lengths of the pipe segments equal to 150 km and 1080 km, and for both NS1 leaks - 230 km and 1000 km, respectively. The system of equations derived in Section 3 is evaluated for these four segments with the parameters summarized in Table 1.

There is an uncertainty about the pressure in the Nord Stream 1 pipelines. The Danish energy authority reported pressures
of 165 bar and 103 bar for NS1 and NS2 lines, respectively (https://twitter.com/Energistyr/status/1576888899288256514). The figure for NS2 agrees well with the data of Sanderson (2022). The figure for NS1 is close to the design pressure of the pipeline (170 bar, http://www.nord-stream.com/en/the-pipeline/facts-figures.html, accessed on 04.11.2011), which is hardly consistent with the statement from the same tweet that the pressure had been lowered in the pipelines by the moment of incident. Since NS1 and NS2 have very similar characteristics, we consider 105 bar as a reliable estimate of the pressure of both pipelines at
the moment of the incident.

The temporal evolution of the pressure at the landfall facilities of the pipelines, calculated for the parameters in Table 1, is given in Fig 5. The figure can be directly compared to the readings of the pressure gauges at the landfall facilities. The plot was

sent to the Nord Stream AG on 16.11.2022 with a request for comments. However, no reply has been received by the moment of the paper submission.

The gas discharge rates, resulting from the solution of the above equations for both pipelines, can be seen in Fig. 6a. Both pipelines exhibit the same starting discharge rate, as it is fully determined by the pipe size and its initial pressure. The NS2A shows a more rapid decrease of the rate and then stabilizes after the shorter part of pipe A (200 km) has been drained. Then the longer end (1000 km) was gradually draining. The breaches of the NS2 pipes occurred closer to the middles of the pipes, causing the initial decrease of the discharge to be slower and the total duration of the discharge to be shorter.

The injection heights for the releases, evaluated with the one-step procedure suggested by Sofiev et al. (2009), are given in Fig. 6b. The initial phase of the release produces a plume that is up to 3.5 km tall, with the height then quickly decreasing down to approximately 1 km.

### 5.3   Validating the injection heights

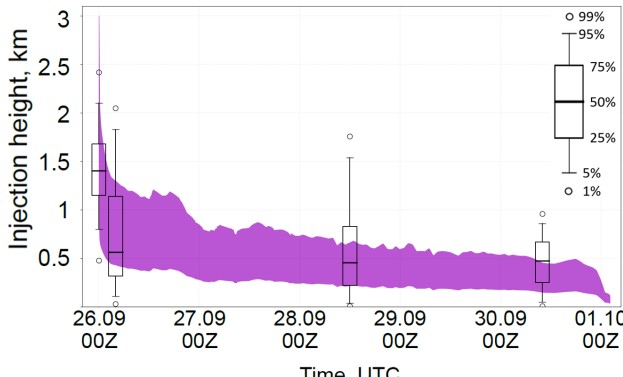

**Figure 7.** Comparison of the plume height parameterisation with UCLALES simulations for NS2A breach. Purple – parameterisation, boxes – LES simulations.

    To ensure the applicability of the parameterisation for fire plumes (Sec. 4) to the methane releases, we simulated the rise

of the buoyant plume with the large eddy simulator UCLALES. We selected 4 periods of the NS2A breach to simulate for comparison: the beginning of the release with the maximum release rate and the moments when the release rate had reduced to 1000, 100 and $10\,\mathrm{kg\,s^{-1}}$ (9.26 00:00, 9.26 4:00, 9.28 12:00 and 9.30 10:00 respectively). The LES simulations were initialized with meteorological profiles from ECMWF forecasts and were allowed to run until the methane tracer crossed the domain downwind boundary. The plume rise was assumed to be complete at the spot downwind where the vertical wind component no

longer correlated with the methane mixing ratio. The height of the plume between that spot and a location 15 km downwind from the release was analyzed.

    Fig 7 shows the comparison of the LES simulated plume heights (box plots) with the fire plume parameterisation (purple). The plume heights computed by the two methods agree reasonably well. In both cases only the initial release peak is strong

enough to inject most of the methane into the free troposphere above the boundary layer (which was 664 meters thick according to ECMWF forecast). For the later and weaker releases, the LES predicts a part of the methane reaching much higher altitudes than given by the parameterisation. However, the models agree that majority of methane stays within the boundary layer (902 m, 490 m and 680 m for the second, third and fourth case) and there is a significant overlap in the region where most of the plume is located. The disagreement in the lower part comes from the LES freely mixing the methane through the boundary layer, while the parameterisation assumes a fixed plume bottom located at 1/3 of the top height. The observed differences do not validate major changes of the large scale model, as boundary layer mixing will occur at the lower end of the plume in a limited amount of time. Thus, the skill of the fire plume parameterisation seems sufficient for predicting the rise of buoyant gas plumes.

The width of the emission area of the gas is an uncertainty of the LES simulations. In the main simulations, the release is assumed to be distributed normally within a circle with a diameter of 500 m. We conducted sensitivity studies varying the diameter from 100 to 1000 m for the highest release case. We found a very limited sensitivity to this parameter - while the narrow emission produces a somewhat narrower plume, the mean height of the plume stays practically the same (see Supplementary Material).

## 6  Simulating the methane dispersion from the NS leaks

Using the emission source defined in the above sections, we simulated the methane dispersion for ten days following the release start with the three setups described in Sec 2.1. For each setup, besides the emission sources with plume rise, we have used several fixed vertical profiles to evaluate the sensitivity of the simulations to the injection height.

For each resolution, a set of vertical injection profiles were simulated: 0–50 m, 0-500 m, 0–1500 m, 0-5000 m, and a dynamic vertical profile, labelled as 'FRP' and described in Section 4. The latter injects uniformly into an elevated layer with bounds controlled by the buoyancy flux and meteorological conditions. In all simulations the same temporal profiles of net emission were used.

According to the simulations, during the period from 26.09.2022 to 05.10.2022, the methane plume hit several of the ICOS stations that were actively reporting data. For most stations, the observed variations of methane were well within the range of usual variability of the methane mixing ratios, so one can not unequivocally detect the signal originating from Nord Stream solely from the observed time series. However, if plotted in the same scale as the modelled time series, the peaks originating from the Nord Stream leaks can be relatively well identified.

To illustrate the results of the simulations, we have selected six stations with clearly visible signals. Fig. 8 shows four panels of time series for each station illustrating the observed methane content and the modelled methane excess from the three simulations. The simulations did not have any background methane. Wherever possible, we kept the same vertical scale among the three panels (all stations in Fig 8 except for EE-SMR). The time series for the remaining ICOS and two non-ICOS methane-monitoring sites in Finland and Estonia can be found in the Supplementary material.

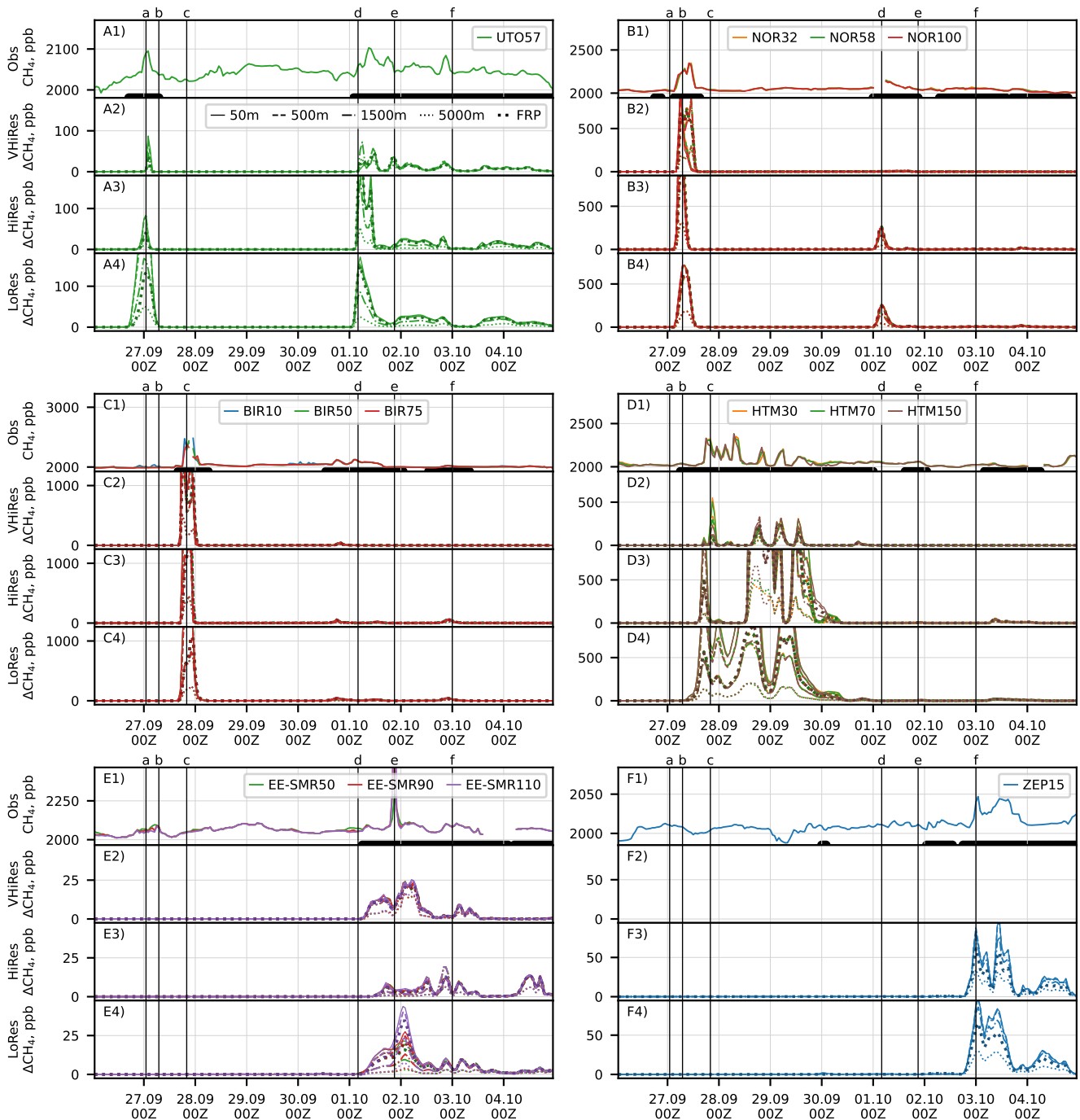

**Figure 8.** Timeseries of methane mixing ratio observed at six selected stations after the pipeline rupture, and corresponding timeseries simulated with three different resolutions for several vertical profiles of the release. Each group of panels corresponds to a station. The panels in each group are (top-down) for observations, and model with $0.02°$, $0.1°$, $0.4°$ resolution. Measurement heights are coded with colours, and emission heights are with line styles. Vertical lines mark the moments shown in Figs. 9–11

. Periods, used to evaluate the model scores for each station are given at the bottom of OBS panels.

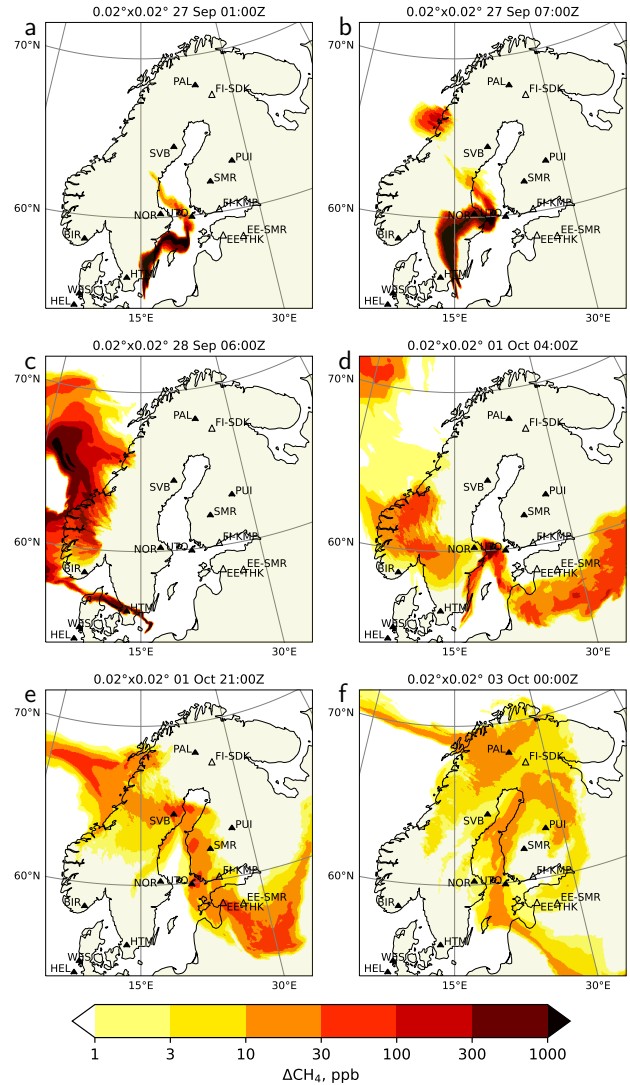

**Figure 9.** Snapshots of near-surface methane excess simulated at $0.02°$ resolution with SILAM driven with Harmonie meteorological fields for FRP injection profile (VHiRes setup). The ICOS stations are shown with filled symbols and three-letter codes, and other stations have two-letter country prefixes. Full list of the station data and references to them can be found from supplementary materials. The panels correspond to the moments marked with vertical lines in Fig. 8.

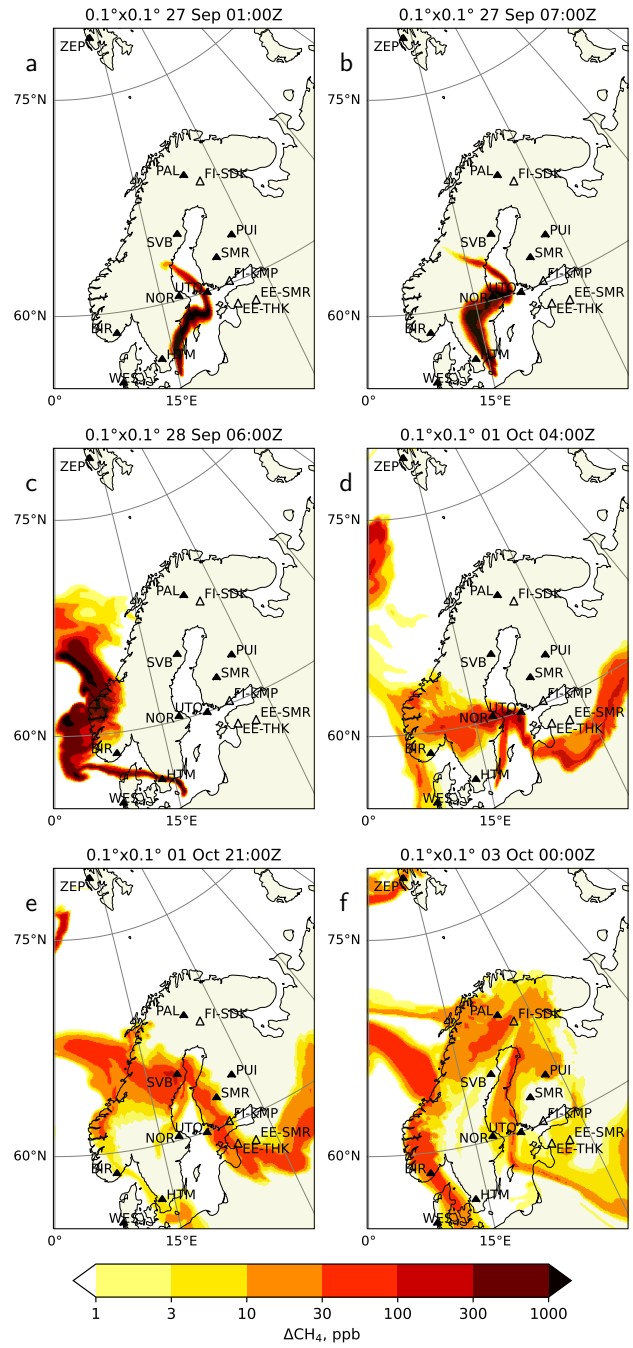

**Figure 10.** Same as in Fig. 9 but for $0.1°$ simulation driven with IFS meteorology (HiRes setup)

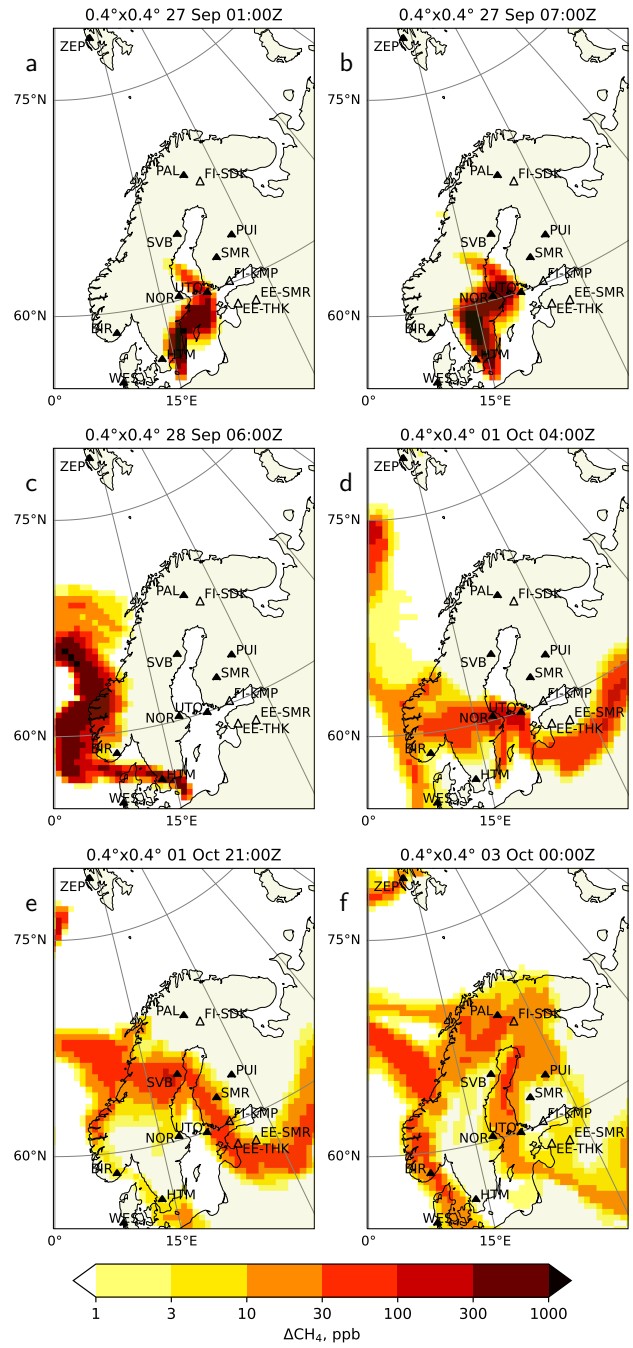

**Figure 11.** Same as in Fig. 9 but for $0.4°$ simulation driven with IFS meteorology (LoRes setup)

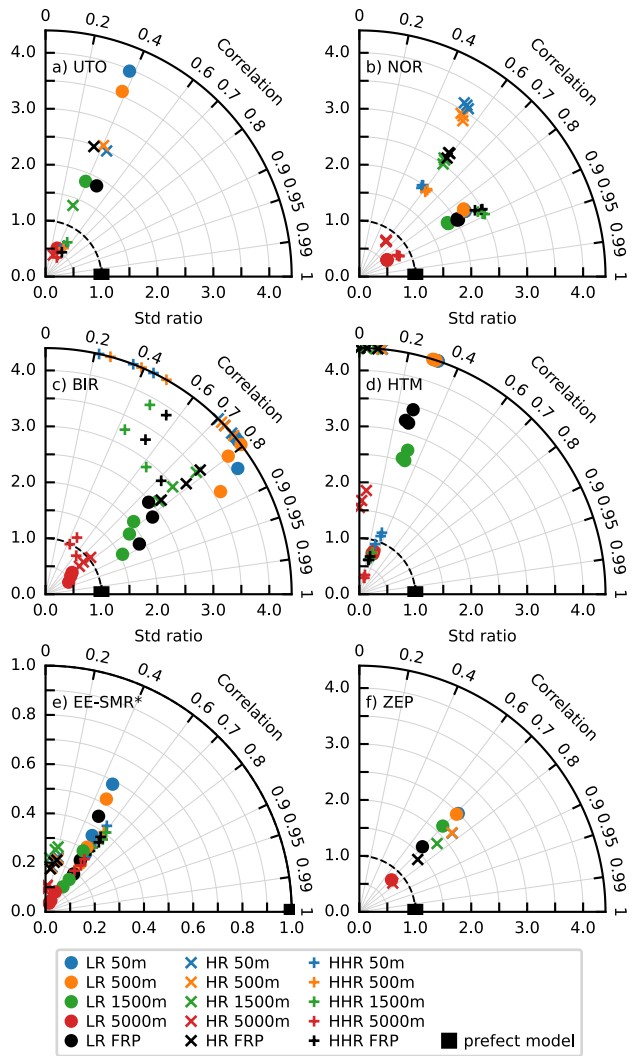

**Figure 12.** Taylor diagrams of model performance on the timeseries shown in Fig. 8. An ideal model is given by a back rectangle. Note the excluded observed peak for the EE-SMR scores in Panel e). The scores beyond the plot range shown at the corresponding edge of a panel.

Six moments of time were selected to illustrate the spatial distribution of the simulated methane plume and its relative position to the stations. The selected moments are marked with vertical lines and letters at the top of the panels with observations in Fig 8. The maps of near-surface methane mixing ratios for the selected moments are shown in the corresponding panels in Fig. 9, Fig. 10, and Fig. 11.

Direct quantitative model evaluation of the performed simulations against the observational data poses a certain difficulty, since one has to compare observed methane levels to the simulated excess methane. In the case of a large excess the background variations can be neglected. On the other hand, the highest concentrations are also the most uncertain ones, since the plumes are relatively narrow, and the time series of both observations and simulations are determined way more by fine details of plume location than by emission rates. Due to the small spatial extent of the source, minor inaccuracies of the dispersion model and/or driving meteorology can lead to significant discrepancies between the model and the observations even for perfectly accurate emission profiles.

For the sake of completeness, we have performed a quantitative evaluation of the simulation results. To allow for a spatially inhomogeneous background as well as for temporal variability, we used metrics that do not depend on the model bias, i.e. correlation, ratio of standard deviations (STD ratio) and normalized de-biased root-mean-square error (RMSE). These quantities can be naturally represented by Taylor diagrams (Taylor, 2001), where the de-biased root-mean-square error normalized with the STD of observations appears as a distance from the "perfect" model. For evaluation, we selected periods when any of the simulations predicted an observable methane excess, assumed to be at least 1 ppb. The selected periods for each station are marked in the corresponding time series panels (Fig 8. The Taylor diagrams for the selected stations are shown in Fig. 12, whereas the diagrams for the rest of the stations can be found in the Supplementary material. Since the magnitude of the variability of both the excess methane content and the background methane vary a lot among the stations, we could not find a way to aggregate the scores among the stations in a reasonable manner to make a solid conclusion on the relative performance of the simulations.

In the diagrams of Fig. 12, the Pearson correlation coefficient is represented by the angle with the y-axis and the ratio of standard deviations by the distance from the origin. The de-biased root-mean-square error, normalized with the standard deviation of the observations, equals then the distance from the "perfect model". The shapes of the markers refer to the different model setups, while the emission injection heights are coded by color. Different observation heights are thus shown with markers of the same shape and color. The markers plotted along the correlation axes of the figures represent values that don't fit inside the plotted areas of figures.

The earliest detection of the plume occurred at the Utö station (UTO), located in the Baltic sea (Fig 8Ax), around midnight 27.09.2022 (Figs 9a, 10a, and 11a). The timing of the peak is in good agreement between the observation and all simulations. In all simulations, the plume touched the station without crossing it, therefore the magnitude of the peak both in observations and simulations was strongly influenced by fine details of the plume location. The VHires and HiRes simulations produce narrower peaks than LoRes ones. The magnitude of the peak for HiRes simulations was well reproduced for the fixed injection heights in the range of 500-1500 meters, and with the dynamic injection profile. LoRes simulations clearly overestimate the peak, especially for lower injection height (reached 350 ppb). The peak originates from early-stage high-altitude injection.

There is also a good correspondence between the measurement and the modelled evolution of the time series during 01.10.2022 and 02.10.2022, except for a large peak during the first half of 01.10.2022 in lower-resolution simulations. During that time the station was at the edge of the plume (see Figs 9b, 10b, and 11b), where slight uncertainties of the plume location lead to large differences in simulated concentrations. The resulting correlation is in the range of 0.3 – 0.6, with the highest correlation for the VHires case, which is also the least sensitive to the injection height (Fig 12a). At the same time the standard-deviation ratio is around 0.5 – 0.7 for these cases. For other cases the effect of the injection height is stronger.

The Norunda station (NOR) in Sweden (Fig 8Bx) measures methane at three different heights, all of which reported very similar mixing ratios during the simulated period. In the morning of 27.09.2022, the arrival of the plume at the station resulted in a clear increase of methane, i.e. by about 300 ppb (see Figs 9b, 10b, and 11b). The simulations show a notably higher peak, of up to 3500 ppb, for the near-surface emission scenario in HiRes case, while the peak magnitude for the 5-km injection height has about correct magnitude. In the VHires simulation both the shape and the magnitude of the observed peak were best reproduced with the fixed 0–5000m injection profile. This indicates that the FRP injection height could be too low at the beginning of the releases. There is a gap in the measurement data corresponding to the arrival of the second peak (01.10.2022 04Z), probably caused by the overly conservative automated quality control of the observational data. The second peak is not visible in the VHires simulation, since the simulated plume was at a higher elevation (see Figs 9d, 10d, and 11d). The timings of both peaks were nicely captured by the model. The highest correlation is shown by the VHires model, except for the scenarios of near-surface injection, and by the LoRes model (Fig 12b). The HiRes case resulted in a too narrow peak. The probable reason is that in LoRes case, a too fast passage of the plume over the station was compensated by excessive smearing of the plume by the low-resolving model. The magnitude of the simulated peaks affects the standard deviation ratio, as it is underestimated for the emission profile of 0–5000 m and overestimated in the other cases.

The Birkenes station (BIR) in Norway (Fig 8Cx) detected a major peak just before midnight 28.09.2022. Similar to NOR, there is a gap in the observations during the peak. The peak simulated with the FRP plume rise model (1300 ppb for VHires and HiRes, and 800 ppb on LoRes) is stronger than the measured one, showing that the injection height was slightly underestimated, again pointing to a too conservative injection height of the FRP plume rise model. The resolution of the simulation did not have a major effect on the peak timing, since the plume was already wide enough when it passed over the station (see Figs 9c, 10c, and 11c). The correct timing of the peaks resulted in a high correlation, i.e. up to 0.9 (Fig 12c), whereas the overestimation of the main peak and the likely lack of measurements of the highest values resulted in a very high STD ratio for all simulations, except for those with the emission extending to 5000-m. Remarkably, there is a large scatter between observations made at different heights, originating from differences in the sampling of the main peak. The secondary peaks occurring after 30.09.2022 have also been reproduced by the model, although they do not affect the performance metrics.

During two days starting from ∼12Z 27.09.2022, the plume was meandering near the Hyltemossa ICOS station (HTM) in Sweden (Fig 8Dx), which resulted in an oscillating pattern in the observations. The plume was narrow (see Figs 9c, 10c, and 11c), so a slight change of the wind direction was able to result in a large change of the methane concentrations at the station. The magnitude and timing of some of the observed peaks were nicely captured by the VHires simulation, since it was able to simulate sufficiently narrow plumes, and its driving Harmonie meteorology reproduced the land-sea circulation well. For

coarser resolutions the simulated concentrations reached up to 5000 ppb for the near-surface emission scenario. Similarly to the metrics for the NOR station, the HiRes simulation exhibits a lower correlation than the others (Fig 12d), since while it is capable of creating finer features of the plume, it fails to reproduce the timings and magnitudes of the peaks. The lower-resolution simulation (LoRes) improves the evaluation metrics by smearing out the plume. The VHires simulation, besides

exhibiting a higher correlation, also resulted in a STD ratio within 50 % of unity for all injection heights except for 5000 m.

The strongest peak of the whole measurement dataset was observed at the Estonian SMEAR station (EE-SMR) around 21Z 01.10.2022 (Fig 8E1). The peak showed a strong vertical gradient of methane, ranging from about 200 ppb of excess methane at 50 m above the ground to 1500 ppb at 110 m. The simulations indicate the arrival of a plume around the same time (Fig 8), but of much wider extent and of about 30-100 times lower intensity. The peak in the LoRes simulation showed a similar vertical

profile of the excess methane content, i.e. the 110 m level exhibited some 50 % higher values than the 90 m one.

The corresponding maps (Figs 9e, 10e, and 11e) show a plume in the vicinity of the station, although with concentrations not exceeding 100 ppb. If we used only he aforementioned threshold of 1 ppb for this station the model points would have collapsed to the origin of the Taylor diagram, due to the failure of the models reproduce the observed peak. To explore the situation further, we excluded the peak (>2150 ppb) from the selection for calculating the evaluation metrics (Fig 12e). With

400 such a selection, the model time series indicates a correlation above 0.6, though with several times smaller mixing ratios than indicated by the observations. This is not necessarily an indication of a corresponding underestimation of the modelled emission, since the background methane at the EE-SMR station exhibits a variation whose magnitude is similar to the modelled variations. Substantial variability of the background at the station leads to the reduction of the STD ratio. Similar to the other stations, the best evaluation metrics are shown by the VHires simulations, while the LoRes simulations correspond to slightly

lower correlations with a larger scatter between STD ratios. The lack of a peak in the simulations indicates that it is likely not originating from the Nord-Stream leaks.

The arrival of the plume at the Zeppelin station (ZEP) at Spitsbergen archipelago was well reproduced by the model (Fig 8F), except for the VHires simulation, that did not extend to the station. The resolution of the simulation had a moderate impact on the magnitude of the methane excess around the station (Figs10f, and 11f), and the dual-peak structure of the time series was

410 well reproduced at both resolutions. In both cases the magnitude of the plume was slightly overestimated with the FRP model and underestimated with the emissions extending to 5-km. The correlations between the model and the observations (Fig 12f) reached 0.75 and all standard deviation ratios were within a factor of 2.5 from unity. Contrary to the other stations, the HiRes simulation shows somewhat better correlations and lower STD ratios than the LoRes one.

## 7   Conclusions

By relying solely on publicly available media reports, we successfully inferred the temporal evolution and the injection height of the Nord Stream gas leaks in September 2022. The resulting inventory specifies locations, vertical distributions and temporal profiles of the methane sources, and can be used to simulate the event with atmospheric transport models. The inventory is supplemented with a set of observational data tailored to evaluate the results of the simulations.

Unlike in many cases of industrial accidents, the total released amount of tracer was relatively well known, amounting to 110 kt of methane in each of the three breached pipes, or 330 kt in total. The main uncertainties stem from the assumption of gas composition in the pipes and the assumption of a 105 bar initial pressure in all damaged pipes. However, we believe that these figures are accurate within a few percent. Since the uncertainties of atmospheric dispersion models are much larger than small uncertainties in the emitted amount, the assumption of the emission consisting of pure methane does not impact the quality of the simulation.

The nature of a pollutant-transport problem with point sources and point receptors causes accurate prediction of the observation results to be unlikely to succeed. A transport model, even if it was perfect on its own, acts as an integrator of errors of the driving meteorological model. Our evaluation of the performed simulations with different setups indicates that a slight change of the plume location and/or shape, caused by uncertainties of the dispersion model or in the driving meteorological data, can lead to huge changes in the simulated time series at the measurement station. The time series are also substantially affected by the spatial resolution of the transport model. This sensitivity has to be accounted for also in inverse problems, where a slight variation of the model setup can substantially change the inversion results.

Even with an accurate temporal profile, the effective injection height of the buoyant admixture has to be parameterised. In this study, we used an existing parameterisation for wildfire plume injection height. However, there is a substantial difference in the mechanisms of buoyancy loss between an overheated moist plume from a fire and a methane plume. The fire plume loses buoyancy due to dilution, stable temperature stratification of the surrounding air and radiative cooling, and gains buoyancy from the latent heat of water vapour condensation, whereas only dilution is relevant for the methane plume. Therefore a specially tailored plume-rise model would be more appropriate for the case. Nevertheless, the fire plume model was able to provide an evolution of the effective injection height for the methane plume that agrees with process-based LES simulations.

In order to reliably compare a simulated plume with regular methane observations, the plume should induce a significant change of the observed times series, i.e. the increment caused by the plume should be larger than the normal variability of the methane concentration at the station. Moreover, in order to rank model setups by simulation quality, the difference between the simulations should be larger than the background variability. As seen from the time series shown in Fig 8 and in the Supplementary material, this was rarely the case in this study. However, by selecting only the times when the model results indicated that the stations were under the influence of the plume originating from the broken pipes, we found that correlation coefficients between observations and model simulations, while rarely above 0.8, exceeded 0.4 for the majority of the stations, even though the model simulations represented only a part of methane variation at a station.

As seen from the plotted time series, the vertical profile of the release influences mostly the height of the peak values of methane through the initial dilution of the plume, while the timing and shape of the modelled peaks are not very sensitive to the release height. Thus the correlation coefficients are often relatively similar for simulations differing only by the release height. As seen from the Taylor diagrams, the lower the correlation, the more the model needs to underestimate the amplitude of the observed variations (standard deviation ratio < 1) to reach the lowest de-biased root-mean-square error. E.g. for near zero or negative correlations, the lowest RMSE is reached by the model setup with the lowest plume concentration. This feature makes it useless to rank the studied model setups by RMSE in order to find the optimal release height. This should also be recognized

when making top-down emission assessments that rely on minimizing RMSE, and we cannot point out the release height that would lead to the best fit with observations due to the large amplitude of the background variations. The concentrations at the stations simulated with different vertical emission profiles differ by factors of several times, which would lead to very different emission inversions depending on the estimated emission profile. The temporal variations of the vertical injection profile should be accounted for when using the case for evaluating source-inversion techniques.

The evaluation of the simulation results against the station data and Large-eddy simulations suggests that the fire-plume injection profile was likely too low for the methane plume, having a low emission rate. However, a relatively small difference in the evaluation metrics between the 50 and 500-m injection heights suggests that this uncertainty had little effect on the long-range transport.

The performed evaluation of the dispersion simulations with several model setups indicated the applicability of the developed inventory to the forward simulations. The inventory together with the ICOS observation data can be used to test and validate various source inversion techniques. The inventory can also be used as a starting point for inversions of the effective vertical and temporal profiles of the plume injection based on column-integrated observations from methane-observing satellites, such as IASI.

*Code and data availability.* The code of SILAM model that can be used to reproduce the results of the current study is available from GitHub https://github.com/fmidev/silam-model/tree/v5_8_2 (Kouznetsov, 2023). Appendix also contains a code to simulate methane leak from a pressurized pipe. The source estimates for the leak at 10-minute resolution together with the evaluated injection heights both in CSV format, and in Silam point-source format are available from the supplementary material. The summary of the observed and simulated station timeseries can be found in supplementary material.

*Video supplement.* The animation of the methane plumes simulated with $0.02°$ resolution can be found in (Kouznetsov and Kadantsev, 2023)

*Author contributions.* RK conceptualised the paper, performed the numerical simulations and evaluations, wrote the initial text and prepared the figures. MS contributed to the case conceptualization, participated in writing and editing of the manuscript. EK prepared the animation of the simulation results. MP adapted UCLALES for this study, performed the LES simulations and participated in analyzing their results, contributed to SILAM development, conceptualisation of the study, and editing the manuscript. RH and AU participated in SILAM development, conceptualisation of the study, and editing the manuscript. YF assisted with literature overview, data mining and editing the manuscript. DK contributed to the design of the conceptual model and to calculations for the gas leak. SN performed the measurements, provided the data for EE-SMR station and contributed to editing the manuscript. HJ performed the measurements and provided the data for EE-THK station.

*Competing interests.* The authors declare that no competing interests are present.

*Disclaimer.* The paper represents the authors' personal opinions and views, which might or might not agree with the positions of their organizations.

*Acknowledgements.* The study was funded by the EU Horizon project EXHAUSTION (grant 820655), FirEUrisk (grant 101003890) and
NKS-B SOCHAOTIC project (contract AFT/B(22)1). Support of the Estonian Research Infrastructures Roadmap project Estonian Environmental Observatory (3.2.0304.11-0395), the Estonian Environmental Investment Support of Academy of Finland project 322532 (MOAC) for the LES modelling is acknowledged. Centre (KIK, grant no. 3-2.8/6574), the Estonian Research Council (project PRG1674) and the European Union's Horizon 2020 Research And Innovation programme (grant agreement no. 871115) ACTRIS IMP is kindly acknowledged.

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
