# Peer review of "A bottom-up emission estimate for the 2022 Nord Stream gas leak: derivation, simulations and evaluation"

_EGUsphere, 2023_

## Referee Comment (RC1)

Review for egusphere-2023-732.

Main comments:

Methane leaks from the Nord Stream pipeline explosion have gained worldwide attention. Although a number of studies or reports have given estimates of methane leakage from this event, detailed data on the rate of methane leakage over time from this event are lacking. This study reconstructs vertical profiles and temporal evolution of the methane releases from the broken pipes, and simulated subsequent transport of the released methane in the atmosphere. The results show that the emission injection vertical profile and the meteorology used to drive CTM can significantly affect the methane dispersion simulations, and which in turn can seriously affect the observation-based emission inversion. Therefore, the data and the results reported by this study are important. The authors illustrate the reliability of the data by comparing the simulations with the observed increase in $CH_4$ concentration, however, the article does not give a clear conclusion on how much the concentrations modelled using the emission rates and vertical profiles given in this paper deviate from the actual observations. In addition, the writing is slightly haphazard, with some writing and formatting problems. Therefore, I would suggest that the article needs to go through a major reversion.

Specific comments:

1. Abstract, Don't divide it into so many paragraphs.

2. Lines 33-35, Please check the units, they need to be the same when comparing.

3. Line 35, what's the mean of 60%?

4. Line 43, 99, Problems with reference citation format.

5. Line 94, which product of ECMWF forecasts was used? ERA5?

6. Section 2.1, the setup of SILAM model should be added in this section, including the simulation domains, horizontal resolution, vertical layers, etc.

7. Lines 109-115, UCLALES simulations were driven using the ECMWF forecasts, however, UCLALES was run with 50 m horizontal and 10 m vertical resolution, while the forecasts are in 0.1×0.1 degrees. Does this severe mismatch in spatial resolution affect the simulation results? Why not use MEPS? It has much higher resolution of 2.5 km.

8. Line 109, Writing problems. "with temperature, humidity and wind profiles and surface variables" should be changed to "with temperature, humidity, wind profiles, and surface variables".

9. Line 118, Writing problems. "…Observation System (ICOS) network, https://icos-cp.eu (accessed 30.11.2022)." should be changed to "Observation System (ICOS) network (https://icos-cp.eu, accessed 30.11.2022)."

10. Line 139, full name of "rhs".

11. Line 143, Formatting issues, "$4 \cdot 10^7$" should be changed to "$4 \times 10^7$".

12. Line 155 and the other places, Formatting of units throughout the text needs to be standardized. According to the formatting requirements of ACP, units must be written exponentially (e.g. W m$^{-2}$).

13. Table 1, Please use a 3-wire table. Horizontal lines should normally only appear above and below the table, and as a separator between the head and the main body of the table.

14. Line 260-264, It is suggested that this paragraph add a description of the total estimated leakage and a comparative analysis with existing results, rather than just describing changes in emission rates over time.

15. Lines 266-267, as shown in Figure 6, it is better to use the values of 3.0 and 0.5 km, rather than 3.5 and 1.0 km.

16. Line 275, Why is it 15km here?

17. Line 302 and the other places, problems of the date format.

18. Lines 302-306, Here, the author needs to give some descriptions about the differences between the simulation results with different vertical profiles and different model resolutions, and it is also recommended to give some quantitative analyses to clarify which vertical profile and model resolution simulation results are in better agreement with the observations.

19. Lines 309-310, remove the sentences of "Colors correspond to observation heights for each station. The line style shows different vertical distribution of the emission." For a description of the elements in the diagram, just put it in the diagram title.

20. Figure 8, It is difficult to distinguish too many lines in the same plot, it is suggested that 1) the same site, with different heights, should be treated as different sites instead of being drawn in the same plot; 2) different colours should be used to represent different vertical profiles in the simulation results instead of using different line types; 3) the observed data can be plotted after removing the background so as to make better comparisons with the model.

21. Lines 316-317, the same as comment 19, remove the sentences of "The ICOS stations are shown with filled symbols and three-letter codes, and other stations have two-letter country prefixes. Full list of the station data and references to them can be found from supplementary materials."

22. Line 375, Incomplete sentences.

23. Line 375, "3$x$95 kt" should be changed to "3×95 kt" or "285 kt".

---

## Referee Comment (RC2)

This manuscript describes a bottom-up estimation of methane emissions caused by a gas leak in the Nord Stream pipeline system. Given the methane's climate-forcing aspect and current geopolitical situation, it addresses a relevant topic. It can contribute to advancing the field of emergency management for large-scale events, particularly for atmospheric releases not yet described in scientific literature. The approach makes use of available concepts and tools to provide emission estimations and set up a dataset for evaluating atmospheric dispersion simulations for assessing the impact of such releases.

General comments:

The manuscript is well-written and thoroughly describes the steps and caveats of estimating the methane emissions leaking from the pipes. However, the discussion needs to compare the emission estimates by the method described and other estimations available in the literature, such as the referred Sanderson (2022) and other work not referred to in the text, such as Jia et al. (2022). The conclusions are vague, not providing the values obtained when applying this methodology and how much it deviates from the observations used to validate this study.

Some aspects needing consideration are described in the specific comments below.

Specific comments:

- Section 2.1 needs to include the SILAM model's spatial (including vertical) and temporal resolution. The different set-ups of the model are described in Section 6, but it may be worth describing it here instead.
- Section 2.3 When describing the stations, add the abbreviation to be shown in the figures, etc., making it easier to identify the stations. The authors should use these abbreviations in Section 6.
- Section 3, Figure 2 presents similar colours for 2m and 96 h.
- Section 6:
  - Figure 8 has much valuable information in a very condensed way. However, it can be challenging to distinguish the deltas from the different injection heights, making it hard to understand the performance of FRP vs prescribed heights. The Figure caption is missing something at the end.
  - Add the short name of the model set-up to the captions of Figures 9-11.
  - line 296: is the term cloud correct here?
  - line 299: "in the morning 29.9", do you mean 27.9?
  - line 319: "excess methane at 30 m", do you mean 50 m?
  - How much will the difference in the spatial domain in VHires affect the recirculation of the plumes?
- Section 7:
  - Why 3x95,000 tons and not 285 000 or 285kT?
  -

Reference: Jia et al (2022) https://www.sciencedirect.com/science/article/pii/S2666498422000667

---

## Author Response (AR1)

**Responses to the comments on a discussion paper "A bottom-up emission estimate for the 2022 Nord Stream gas leak: derivation, simulations and evaluation"**

Rostislav Kouznetsov[1], Risto Hänninen[1], Andreas Uppstu[1], Evgeny Kadantsev[1], Yalda Fatahi[1], Marje Prank[1], Dmitrii Kouznetsov[2], Steffen Noe[3], Heikki Junninen[4], and Mikhail Sofiev[1]

[1]Finnish Meteorological Institute, Helsinki, Finland
[2]University of Electro-Communications, Tokyo, Japan
[3]Estonian University of Life Sciences, Tartu, Estonia
[4]University of Tartu, Estonia

**Correspondence:** Rostislav Kouznetsov (Rostislav.Kouznetsov@fmi.fi)

We would like to thank both reviewers for their valuable comments. Below is the detailed response to the critical points. The reviewer's statements are given in *italic*.

**1 Response to the comments from the reviewer 1 (RC1)**

**1.1 Main comments**

5 *... however, the article does not give a clear conclusion on how much the concentrations modelled using the emission rates and vertical profiles given in this paper deviate from the actual observations.*

**Response:** We have added a quantitative evaluation of the simulations against the observational data with Taylor diagrams. Since the stations are not too numerous and rather different we could not find any reasonable way to aggregate the scores to a single number per simulation, so the evaluation has been done at per-station level.

10 *In addition, the writing is slightly haphazard, with some writing and formatting problems.*
**Response:** We have improved the text.

**1.2 Specific comments**

*1. Abstract, Don't divide it into so many paragraphs.*
**Response:** The paragraphs merged, abstract shortened.

15 *2. Lines 33-35, Please check the units, they need to be the same when comparing.*
**Response:** The units corrected. They are now Tg for a single year or event, and Tg/y for rates.

*3. Line 35, what's the mean of 60%?*
**Response:** We rephrased the statement.

*4. Line 43, 99, Problems with reference citation format.*

**Response:** Fixed

*5. Line 94, which product of ECMWF forecasts was used? ERA5?*

**Response:** It was high-resolution operational global forecasts (HRES product). Clarification added

*6. Section 2.1, the setup of SILAM model should be added in this section, including the simulation domains, horizontal resolution, vertical layers, etc.*

**Response:** The simulation setup moved to section 2.1, descriptions of the domains added

*7. Lines 109-115, UCLALES simulations were driven using the ECMWF forecasts, however, UCLALES was run with 50 m horizontal and 10 m vertical resolution, while the forecasts are in 0.1×0.1 degrees. Does this severe mismatch in spatial resolution affect the simulation results?*

**Response:** UCLALES computes its own dynamics, so it is not driven by ECMWF or any other weather model. Rather, the ECMWF forecast data was used to initialize the UCLALES simulations. UCLALES is initialized by single value for surface variables and single profile of temperature, wind, moisture etc. for the whole horizontal domain. The profiles are interpolated to the UCLALES levels. Random perturbations are added to these values and turbulence is allowed to develop during the model spin-up. As the size of the UCLALES domain is comparable to the ECMWF grid cell, it is not inappropriate to use this data as initialization. How different the results would be using MEPS data depends on how much the MEPS profiles differ from ECMWF ones, however, it cannot be claimed that they would be more appropriate to use than the ECMWF ones to describe the state in the whole UCLALES domain.

*8. Line 109, Writing problems. "with temperature, humidity and wind profiles and surface variables" should be changed to "with temperature, humidity, wind profiles, and surface variables".*

**Response:** Fixed

*9. Line 118, Writing problems. "… Observation System (ICOS) network, https://icos-cp.eu (accessed 30.11.2022)." should be changed to "Observation System (ICOS) network (https://icos-cp.eu, accessed 30.11.2022)."*

**Response:** Fixed

*10. Line 139, full name of "rhs".*

**Response:** Fixed

*11. Line 143, Formatting issues, "4·107" should be changed to "4×107".*

**Response:** Fixed

*12. Line 155 and the other places, Formatting of units throughout the text needs to be standardized. According to the formatting requirements of ACP, units must be written exponentially (e.g. W m–2).*

**Response:** Fixed

*13. Table 1, Please use a 3-wire table. Horizontal lines should normally only appear above and below the table, and as a separator between the head and the main body of the table.*

**Response:** Fixed

*14. Line 260-264, It is suggested that this paragraph add a description of the total estimated leakage and a comparative analysis with existing results, rather than just describing changes in emission rates over time.*

**Response:**

*15. Lines 266-267, as shown in Figure 6, it is better to use the values of 3.0 and 0.5 km, rather than 3.5 and 1.0 km.*

**Response:**

*16. Line 275, Why is it 15km here?*

**Response:** UCALES, like many other large eddy simulators, runs with periodic boundaries, meaning that when hitting the downwind domain border, the plume together with the disturbances it causes to the flow field would enter the domain again from upwind border and start interfering with the plume development above the source. Thus the simulations had to be stopped before such disturbances would influence the plume rise. However, the wind speeds being higher at higher altitudes, the near surface part of the plume had not reached the domain border yet and thus the vertical profile of the plume had not yet stabilized there. 15 km was selected as the place that the plume had reached at all altitudes and the vertical profile had stabilized and the area downwind of that was left out of the analysis

*17. Line 302 and the other places, problems of the date format.*

**Response:** We brought dates to a uniform dd.mm.yyyy format in the paper. Not sure if there are any Copernicus guidelines on date formatting.

*18. Lines 302-306, Here, the author needs to give some descriptions about the differences between the simulation results with different vertical profiles and different model resolutions, and it is also recommended to give some quantitative analyses to clarify which vertical profile and model resolution simulation results are in better agreement with the observations.*

**Response:** See the response to the main comment above.

*19. Lines 309-310, remove the sentences of "Colors correspond to observation heights for each station. The line style shows different vertical distribution of the emission." For a description of the elements in the diagram, just put it in the diagram title.*

**Response:** The diagram caption already had this info, so we just removed the redundant statement from the text.

*20. Figure 8, It is difficult to distinguish too many lines in the same plot, it is suggested that 1) the same site, with different heights, should be treated as different sites instead of being drawn in the same plot; 2) different colours should be used to represent different vertical profiles in the simulation results instead of using different line types; 3) the observed data can be plotted after removing the background so as to make better comparisons with the model.*

**Response:** Thank you for the suggestion. We have tried that representation as well, but it essentially results in very similar plots, since in most cases different heights have very similar concentrations (both in observations and in simulations). The

cases when there are deviation from a uniform vertical profile are way better seen in the present configuration. Hopefully, with Taylor diagrams added, the problem of evaluating specific setups has been solved to some extent.

*21. Lines 316-317, the same as comment 19, remove the sentences of "The ICOS stations are shown with filled symbols and three-letter codes, and other stations have two-letter country prefixes. Full list of the station data and references to them can be found from supplementary materials."*

**Response:** Done

*22. Line 375, Incomplete sentences.*

**Response:** The conclusions have been rewritten.

*23. Line 375, "3x95 kt" should be changed to "3×95 kt" or "285 kt".*

**Response:** Done

**2 Response to the comments from the reviewer 2 (RC2)**

The reviewer's comments largely repeat the comments given on a preprint submission stage, and we believe most of them have been mostly addressed already in the published discussion paper. Therefore we reproduce our responses highlighting with blue color the statements that were added/updated since the previous review and response.

**2.1 General comments**

*The manuscript is well-written and thoroughly describes the steps and caveats of estimating the methane emissions leaking from the pipes. However, the discussion needs to compare the emission estimates by the method described and other estimations available in the literature, such as the referred Sanderson (2022) and other work not referred to in the text, such as Jia et al. (2022).*

**Response:** Thank you! We have been aware of the Jia et al. (2022) study. It was published rather quickly, which affected its quality. The study has numerous factual, methodological and ethical issues, so we preferred to avoid its citation in the initial paper. However, since both reviewers requested it, we added the reference, pointing out some of the issues.

Among the issues with Jia et al. (2022) study not reflected in our manuscript are:

– Missing acknowledgements to ESA for the map, which was borrowed from https://www.esa.int/Applications/Observing_ the_Earth/Satellites_detect_methane_plume_in_Nord_Stream_leak

– Sentinel-1 SAR backscattering map presented as just "remote sensing image" without any indication of the plotted quantity

– missing color bars at Fig. 2, precluding any comparison of the simulations to observations

– Comparison from their S2 figure suggests much higher uncertainties of the inversion than declared

- Unrealistically low uncertainty and too-good-to-be-true match of atmospheric inversions to the total calculated assuming two pipes broken.

- Unrealistic and unjustified evolution of the release rate needed to fit the observed time series.

*The conclusions are vague, not providing the values obtained when applying this methodology and how much it deviates from the observations used to validate this study.*

**Response:** See the answer to the main point of RC1

**2.2 Specific comments**

*Section 2.1 needs to include the SILAM model's spatial (including vertical) and temporal resolution. The different set-ups of the model are described in Section 6, but it may be worth describing it here instead.*

**Response:** Thank you! A note on the resolutions of SILAM added to Sec 2.1, detailed setups left to the section 6.

*Section 2.3 When describing the stations, add the abbreviation to be shown in the figures, etc., making it easier to identify the stations. The authors should use these abbreviations in Section 6.*

**Response:** Thank you! Abbreviations added to both sections.

*Section 3, Figure 2 presents very similar colours for 2m and 96 h.*

**Response:** Thank you! 96 h color changed.

*Section 6: Figure 8 has much valuable information in a very condensed way. However, it can be challenging to distinguish the deltas from the different injection heights, making it hard to understand the performance of FRP vs prescribed heights. The Figure caption is missing something at the end.*

**Response:** Thank you! The caption has been corrected. Line and marker thicknesses were adjusted, color scheme was unified across panels to approximately match the measurement height. Slightly darker color used for FRP timeseries.

*Section 6: Add the short name of the model set-up to the captions of Figures 9-11.*

**Response:** Fixed. Thank you!

*Section 6: line 296: is the term cloud correct here?*

**Response:** Thank you! Replaced with 'plume'

*Section 6: line 299: "in the morning 29.9", do you mean 27.9?*

**Response:** Yes. Thank you!

*Section 6: line 319: "excess methane at 30 m", do you mean 50 m?*

**Response:** Fixed.

*Section 6: How much will the difference in the spatial domain in VHires affect the recirculation of the plumes?*

**Response:** Not too much in this case.

*Section 7: Why 3x95,000 tons and not 285 000 or 285kT?*

**Response:** Fixed